# Molecular Communication for Coordinated Seed and Fruit Development: What Can We Learn from Auxin and Sugars?

**DOI:** 10.3390/ijms20040936

**Published:** 2019-02-21

**Authors:** Hélène S. Robert

**Affiliations:** Mendel Centre for Genomics and Proteomics of Plants Systems, CEITEC MU-Central European Institute of Technology, Masaryk University, 625 00 Brno, Czech Republic; helene.robert.boisivon@ceitec.muni.cz; Tel.: +420-549-49-8421

**Keywords:** auxin, sucrose, embryo, embryo, endosperm, seed, fruit, molecular communication

## Abstract

Seed development in flowering plants is a critical part of plant life for successful reproduction. The formation of viable seeds requires the synchronous growth and development of the fruit and the three seed structures: the embryo, the endosperm, the seed coat. Molecular communication between these tissues is crucial to coordinate these developmental processes. The phytohormone auxin is a significant player in embryo, seed and fruit development. Its regulated local biosynthesis and its cell-to-cell transport capacity make of auxin the perfect candidate as a signaling molecule to coordinate the growth and development of the embryo, endosperm, seed and fruit. Moreover, newly formed seeds need nutrients and form new carbon sink, generating high sugar flow from vegetative tissues to the seeds. This review will discuss how auxin and sugars may be considered as signaling molecules to coordinate seed and fruit development.

## 1. Auxin in Seed Development

Plant hormones are instrumental players for many aspects of plant development. In flowering plants, every step leading to seed formation requires crosstalk between various hormones: flower primordium development, floral organ development, including stamens, gynoecium patterning, ovule formation, ovule number, fertilization, seed formation, fruit initiation [1,2,3,4,5,6,7,8]. To ensure the optimal success of seed production, many steps of seed and fruit development require to be tightly coordinated. Involved in the coordination of the development of these tissues are notably (but not exclusively) plant hormones. At least three hormones are involved in these processes: auxin, gibberellic acid (GA) and cytokinin. Auxin, indole-3-acetic acid (IAA), is produced by a few parallel pathways starting from tryptophan (Trp). The biosynthesis pathway with indole-3-pyruvic acid (IPyA) as an intermediate has been established as the primary source of auxin for plant development [9,10,11]. The enzymatic reactions Trp-IPyA-IAA required two families of proteins: TRYPTOPHAN AMINOTRANSFERASE OF ARABIDOPSIS (TAA1) (also known as WEAK ETHYLENE INSENSITIVE8–WEI8 [12]) and related (TARs), and YUCCA (YUC) flavin-dependent monooxygenases [13]. These enzymes are locally expressed contributing to a local production of auxin [12,14,15,16,17,18,19,20,21]. Auxin is then actively transported cell to cell by auxin influx (AUXIN1 (AUX1) and Like-AUX1 (LAXs)) and auxin efflux (PIN-FORMED (PIN), ATP BINDING CASSETTE subfamily B/P-GLYCOPROTEIN (ABC/PGP)) proteins, whose cellular localization is indicative of the auxin flow direction, creating morphogenic auxin gradients [22,23,24,25]. Discrete accumulation of auxin is sensed by the nuclear receptors TRANSPORT INHIBITOR RESISTANT/AUXIN RESPONSE F-BOX (TIR1/AFB) proteins belonging to the SCF^TIR1/AFB^ complex, activating a transcriptional signaling pathway. Auxin perception triggers the targeting of the AUXIN/INDOLE-3-ACETIC ACID (Aux/IAA) proteins to ubiquitination and proteasomal degradation. This degradation prevents the interaction of the Aux/IAA with the AUXIN RESPONSE FACTOR (ARF) transcription factors. The ARFs specifically bind to DNA on auxin response elements (AuxRE) in the promoter of auxin-responsive genes. Once free from Aux/IAA interaction, ARFs will act either as transcriptional repressors or activators on the target genes in response to the auxin input for an appropriate developmental response [26]. Two types of reporters measure auxin signaling levels. The first one monitors auxin input to the signaling pathway, designed from the ubiquitin-targeted domain of Aux/IAA (domain II, DII). The abundance of DII, coupled with fluorescent reporter genes and expressed under various (constitutive or tissue-specific) promoters, is auxin-dependent, as well as depending on the activities of TIR1/AFB and of the proteasome. The cellular level of DII is negatively correlated to the cellular auxin level. Two main types of these reporters exist, DII and its mutated, auxin-resistant version mDII in separate lines, and R2D2, where these two versions are combined in one transgene and used for cellular radiometric measurements [27,28]. The second reporter, namely DR5, is a transcriptional output reader of the auxin signaling, made of a multimeric AuxRE-based promoter driving reporter genes [22,29]. The activity of this reporter directly correlates with the output of the auxin signaling pathway. Within this review, the role of auxin as a signaling molecule for communication between tissues during seed formation and its influence for fruit set will be discussed.

Seeds are formed upon fertilization of ovules. The mature ovule consists in sporophytic integuments enveloping and protecting an embryo sac containing an egg cell, two synergids, a central cell with two polar nuclei and three antipodal cells, aligned along a micropylar-chalazal axis [30,31]. The egg cell and the two synergids are facing the micropyle, an opening within the integuments for the pollen tube to enter and access the embryo sac [32]. Following fertilization, the egg cell becomes a zygote which elongates and divides asymmetrically into a small apical cell (future proembryo) and a long basal cell, the suspensor. The suspensor forms a cell file connecting the proembryo to the integuments. The apical cell undergoes a series of cell divisions forming a pre-globular embryo of eight cells. This embryo further divides and progresses to the morphogenesis phase with the formation of an embryo epidermis (the protoderm), a root, a shoot and two cotyledons in *Arabidopsis*, leading to a heart-shaped embryo [33]. Subsequently, the embryo matures, and the seed prepares itself for desiccation and dormancy. During the same fertilization process, a second sperm cell will fuse with the central cell, developing into a triploid endosperm. The endosperm, terminal tissue, nurtures and supports the embryo, ensuring nutrient transfer from the sporophytic maternal tissues to the embryo. The endosperm is essential for proper embryo development and seed viability [14]. In parallel to these events, the seed integuments will differentiate into a seed coat [15]. These three seed structures, formed following fertilization, have a coordinated development, despite being of different genetic origins: the seed coat is sporophytic (2n maternal), the embryo is equally maternal (1n) and paternal (1n), and the endosperm is triploid (2n maternal, 1n paternal).

The gynoecium is the female reproductive organ. It is composed of two fused carpels separated by a septum and a transmitting track in *Arabidopsis*. It harbors the ovules. The stigma, recipient of pollen grains, ends the gynoecium’s apical tip. After germination on the stigma, pollen tubes travel through the style and the transmitting track to the ovule for fertilization. The gynoecium becomes a fruit, called silique in *Arabidopsis*, which carries, protects and finally disperses the seeds [34,35].

Auxin biosynthesis, transport and signaling have been implicated in the different steps of the development and morphogenesis of these female and seed structures [7,36]. With local production and discrete signaling, auxin gradients will synchronize the development of the fruit, the seed and the embryo following gynoecium pollination and ovule fertilization. This review will summarize recent findings of how the movement of auxin between these seed structures synchronized their development. In chapter 4, I will additionally shortly discuss the role of a second molecule, sucrose, as a mean of coordinating the development of seed structures.

## 2. Auxin Movement within the Different Seed Structures Triggers Different Developmental Outputs

One of the examples of perfect growth coordination between fruit and seeds is the positive correlation between the number of seeds developed after fertilization and the length of the silique [1]. Current models for fruit set or fruit initiation are based on data suggesting that seeds produce auxin and GA after fertilization, which promotes the growth of the pollinated fruit [1]. This process can be mimicked by application of any of the hormones on the fruit in the absence of pollination, resulting in the formation of seedless (or parthenocarpic [fruit formed without fertilization]) fruit [37]. Therefore, following fertilization, seeds are a source of auxin and auxin is required for the growth of the fruit. Recent publications demonstrated that seed-produced auxin is of importance for the development and growth coordination of the three structures contained in the seed: embryo, endosperm, seed coat [14,15,17,38]. This recent knowledge about inter-tissular communication within the seed and the fruit will be discussed below.

### 2.1. Auxin Movement within the Integuments and Funiculus

Several research groups observed that fertilization of the ovule triggers an increase in auxin levels in the integuments as monitored by the auxin input reporter R2D2 and by analytic quantification [17,38], and the activation of auxin signaling as visualized by DR5-based markers [1,15,17]. Altogether these observations suggest that auxin production increases in ovule integuments following fertilization. Expression pattern analysis of seeds expressing DR5-based reporters [1,15,17] identified that auxin signaling is activated in two distinct regions of the seed integuments, at the micropyle where the embryo connects to the integuments and in the chalazal domain, where the seed is attached via the funiculus to the fruit. Besides physically attaching the seed to the silique, the funiculus plays a role of conduit between these two organs, transporting in both directions hormones, nutrients and sugars. The application of N-1-naphthylphthalamic acid (NPA), a chemical that blocks auxin transport, mimics the fertilization-induced DR5 signals [1]. The DR5 signal at the chalazal region and funiculus overlaps with the vasculature, suggesting that auxin (or auxin-derived signal) might be transported from the ovule to the fruit after pollination [1,38]. Larsson et al. (2017) [38] compared the expression pattern of genes involved in auxin biosynthesis (*YUC*s) and auxin transport (*PIN*s) at different female gametophyte (FG) developmental stages (FG5–7, according to [30]) and during early seed development. The authors observed that (1) auxin accumulation at the chalazal region and funiculus marks vascular precursor cells in ovules at pre-anthesis, (2) an anther—dependent signal reduces polar localization of PIN proteins in the funiculus to limit auxin transport through the funiculus in ovules at anthesis, and (3) xylem tracheary element cell file is extended through the funiculus, and PIN-dependent auxin transport is still restricted after fertilization of the ovule. However, the increase in auxin levels in the integuments following fertilization does not appear to be due to changes in *YUC* expression [38] but rather to enhanced *TAA1* expression levels [17]. Also, the *PIN3* expression domain extends within the integuments following fertilization. Other auxin transport proteins (ABC/PGP1 ABCB/PGP19, AUX1, LAX1) are also expressed in this sporophytic tissue [17,39,40]. Together with the restriction of auxin flow through the funiculus, these could contribute to the higher auxin levels in the integuments following fertilization [17,38]. The biological relevance of these dynamic changes in auxin distribution is that auxin would be a trigger for the formation of the vascular strands within the funiculus, and the synchronous development of the ovule/seed and the gynoecium/fruit, in order to avoid the formation of parthenocarpic fruits.

### 2.2. Auxin Movement from the Endosperm to the Integuments

The initiation of seed development is controlled by epigenetic regulators of the Polycomb group (PcG) family. PcG proteins block the development of the endosperm in the absence of fertilization, by targeting repressive trimethylation on lysine 27 of histone H3 (H3K27me3) at target loci [41]. For female gametophyte and endosperm development, the involved PcG complex is FIS-PRC2 (FERTILIZATION INDEPENDENT SEED-Polycomb Repressive Complex 2), composed of FIS2, MEDEA (MEA), FERTILIZATION INDEPENDENT ENDOSPERM (FIE) and MULTICOPY SUPPRESSOR OF IRA1 (MSI1). The FIS-PRC2 complex is repressing the development of the endosperm prior to fertilization in order to block the formation of fertilization-independent seeds, containing an endosperm but no embryo [41].

The division of the central cell nuclei, triggered by an increase in auxin levels after fertilization, marks the initiation of the endosperm development. Two *YUC* genes are expressed in the endosperm, *YUC10* and *YUC11*. The latter is also present in the central cell before fertilization, co-expressed with one of the auxin-conjugating enzymes GRETCHEN HAGEN3 (GH3.5) [16,38]. Figueiredo et al. (2015) [14] showed that FIS-PRC2 represses the expression of the maternal *YUC10* copy in the central cell before fertilization. The paternal *YUC10* copy brought by the pollen sperm cell is expressed in the fertilized central cell and is necessary for the initiation of the endosperm proliferation. It contributes to the increase in auxin levels in the endosperm as monitored by the R2D2 reporter. In ovules of mutants lacking the FIS-PRC2 function, *YUC10* expression is de-repressed, resulting in an ectopic auxin production (monitored by the R2D2 reporter line) in the central cell without fertilization [14]. Because a fertilization-dependent increase of auxin levels in the central cell is necessary for the proliferation of the endosperm, these observations would explain the autonomous endosperm development in *fis2* and *fie* seeds. Furthermore, the authors identified the MADS-box transcription factor AGAMOUS-LIKE 62 (AGL62) as a signaling component required for this effect. *AGL62* is expressed in the central cell before fertilization and in the endosperm.

Sporophytic-active PRC2 complexes also repress seed coat development before fertilization. A fertilization-derived signal activates seed coat formation by releasing the PRC2 repressing action [41]. It has been demonstrated that auxin is this signal, produced post-fertilization in the endosperm by YUC10, channeled out from the endosperm to the seed coat by AGL62-regulated ABCB/PGP10 auxin efflux proteins [14,15]. Indeed, some seeds (<30%) of mutants in auxin production (*wei8/− tar1/− tar2/+*), but not in auxin signaling (*axr1/+ axl1/−*), are smaller in size, indicating a defect in integument cell expansion. Accordingly, ectopic expression of *TAA1* and *YUC6* in the central cell, prior fertilization, triggers the development of a seed coat. The seeds of *agl62* mutant abort 3 to 4 Days After Pollination (DAP) probably due to an early endosperm cellularization, hypothesized to be the consequence of an absence of development of a seed coat. Integuments of *agl62* seeds are characterized by the absence of auxin and GA signaling responses. The activation of the *DR5* promoter in *agl62* endosperm would suggest that auxin is trapped in the mutant endosperm, consistent with AGL62 being a transcriptional activator of *ABCB/PGP10* expression in the endosperm, and the auxin transport function of ABCB/PGP10 from the endosperm into the integuments [15]. Work from the Köhler lab initiated the characterization of the molecular components involved in the synchronization of the seed coat differentiation with the development of the endosperm following fertilization. They show that it involves the epigenetic regulation of the production and the movement of auxin from the endosperm to the integuments.

### 2.3. Auxin Movement from the Integuments to the Embryo

Another auxin communication flow happens shortly after fertilization between the cells at the micropyle domain and the early embryo. Above we read that auxin level increases in the integuments after fertilization [1,14,17,38]. Robert et al. (2018) [17] observed that *TAA1* expression levels are enhanced in the micropyle region after fertilization, likely contributing to the enhanced auxin levels. Indeed, the presence of DR5-derived signals and the increased levels in auxin metabolites, quantified in fertilized wild-type ovules, are absent in fertilized *wei8 tar1* mutant seeds. Embryos growing in seeds with reduced auxin levels within the integuments (e.g., *wei8*, *wei8 tar1*, *pBAN::iaaL* - a bacterial enzyme degrading auxin, expressed from the *BANYULS* (*BAN*) promoter in the integuments) display morphogenic phenotypes characterized by altered cell division patterns. *TAA1* is not embryonically expressed at this stage. Noteworthy embryos from seeds with reduced auxin signaling in the integuments (*pBAN::BDL*, auxin-resistant version of *aux/IAA12*–*BODENLOSS* gene expressed under *BAN* promoter) are unimpaired despite the absence of a DR5 signal in the micropyle domain. These data, together with reciprocal backcross experiments between wild-type plants and mutant plants with reduced auxin levels, show that the auxin required for establishing a body axis in early embryos is sourced in the micropyle cells upon fertilization [17]. How auxin is transported between sporophytic and embryonic tissues, remain elusive. Expression pattern analysis of several auxin transporter proteins (*PIN*s, *AUX1*, *LAX1*, *ABCB/PGP1* and *ABCB/PGP19*) shows the presence of these proteins in one or both structures, but no explicit demonstration of their involvement in this transport has been presented. AUX1 and LAX1 may contribute to the micropyle-suspensor auxin transport based on their cellular localization [17,39,40]. However, genetic redundancy among other members for the protein family and defects during female gametophytic development complicate the study of sporophytic-embryonic auxin transport. Furthermore, the positional communication between sporophytic tissues facing the embryo sac and the early embryo is conserved between monocotyledonous and dicotyledonous as very similar expression behavior of the DR5 reporter is observed in maize seeds [17].

### 2.4. Auxin Movement from the Endosperm to the Embryo

These studies report the presence of an auxin movement within the integuments, between the endosperm and the integuments after fertilization to promote seed coat development and between the integuments and the early embryo to contribute to the formation of an apical-basal embryonic axis development. Is there any evidence for auxin transport between the endosperm and the embryo? Growth coordination within the seed structures is essential for the viable development of the seed. Therefore, one would suspect auxin to be involved as a signaling molecule for a crosstalk communication between the endosperm and the embryo.

AUX1, ABCB/PGP1 and ABCB/PGP19 auxin transport proteins are localized on zygotic and early embryonic membranes at the interface with the endosperm [17,39,40,42], indicative of a possible auxin transport between the endosperm and the embryo in *Arabidopsis*. Expression of *YUC*s at various developmental stages after fertilization indicates that auxin is produced in the endosperm [14] and embryos [16]. However, there is no concrete evidence that auxin would be transported from the endosperm to the embryo (or vice versa) in *Arabidopsis*.

In maize, auxin levels increase in the endosperm from 8 DAP and remain high until maturation (28 DAP) [43,44]. Auxin plays a positional role for the specification and the development of the aleurone, an outer cell layer of endosperm, separating the seed coat from the starchy endosperm [43,45]. Auxin also accumulates in the ESR (Embryo Surrounding Region) and the BETL (Basal Endosperm Transfer Layer) [43]. The BETL is a specialized endosperm region that facilitates nutrient uptake from the apoplast of the chalazal region. The ESR, as its name indicates, is the layer of endosperm around the embryo. Auxin is produced in maize endosperm, synthesized locally through the IPyA pathway [44]. Expression of *ZmYUC1* and three *ZmTAR* genes were detected, correlating with auxin accumulations [44]. Endosperms of *defective endosperm18* (*de18*) and *defective kernel18* (*dek18*) knockout mutants for *ZmYUC1* have a low auxin content and seed (endosperm and embryo) defect phenotypes [43,44,46]. *ZmPIN1a-c* expression follows the same pattern as *ZmYUC1*, e.g., accumulation in the endosperm following fertilization [43]. In particular, the BETL and the ESR contain high levels of ZmPIN1 proteins after fertilization until the maturation phase. In BETL, ZmPIN1 is localized at membranes facing sporophytic tissues, whereas, in ESR, ZmPIN1 accumulates in the cytoplasm of cells surrounding the embryo suspensor [43]. Additionally, ZmPIN1 is present in the (epidermal) L1 cell layer of the apical region of the embryo, suggesting a possible auxin flow from the endosperm to the embryo in this region [47]. Following this hypothesis, an auxin signaling is detected using a DR5 reporter in the maize endosperm, but not in the embryo, contrary to *Arabidopsis*. This auxin signaling, from the endosperm, may influence the embryonic morphogenesis development in maize [47]. Interestingly no auxin signaling (DR5) signal is observed in the ESR, where PIN1 proteins localize in the cytoplasm rather than at the membranes, which may help to prevent an auxin flow between the ESR and the suspensor cells during early seed development [47]. All together these patterns would create an auxin gradient between the endosperm, the apical embryonic domain and the base of the maize embryo, involved in the directional growth and patterning of the shoot apex of the maize embryo.

In summary, within the seed, three auxin dynamics between the seed structures can be identified and trigger three different developmental responses (Figure 1). In all three cases, auxin production is activated after seed fertilization. Accumulation of auxin in the chalazal region is implicated in the differentiation of the funiculus vascular strands. The second dynamism occurs between endosperm and seed coat. An epigenetic regulation signaling pathway activates auxin production in the endosperm and its transport from the endosperm to the integuments. This transport requires ABCB/PGP10 auxin efflux proteins in the endosperm, transcriptionally regulated by AGL62. Presence of auxin within the integuments initiate its differentiation into the seed coat. The last auxin movement occurs between the micropyle and the early embryo, for the formation of the embryo body axis. Auxin transport requires an intercellular transmembrane transport, relying on transmembrane proteins. Are all cells within the seeds connected and is there any alternative molecular transport means?

## 3. Symplastic Transport, as an Alternative Route of Communication within the Seed

Symplastic molecular movement is an alternative cell-to-cell transport to transcellular transport. Transcellular transport requires the movement of signaling molecules in and out the cell, and across the apoplast, as it is the case for auxin. Symplastic transport is a direct transport between the cytoplasm of two connected cells through a plant-specific structure called plasmodesmata [48]. A symplastic domain consists of a group of cells connected through plasmodesmata. There are several symplastic domains identified in seeds (Figure 2) [49]. Their number and size evolve during seed development. Notably, there is an absence of symplastic linkage between the seed coat and the embryo for the transport of phloem-delivered nutrients from the funiculus. Within the seed coat, three symplastic domains are identified: (1) the symplastic unloading domain at the end of the funiculus in the chalazal domain, linking the developing seed to the silique, (2) the outer integuments and (3) the inner integuments [49]. Until the globular stage, the embryo is one whole symplastic domain. At the globular stage, the symplastic connectivity is restricted between the suspensor and the proembryo, which is then separated into distinct embryonic domains [49]. The symplastic connectivity between the suspensor and the embryo allows the movement of small molecules, including sucrose, hormones and small proteins as BDL/IAA12 [49,50,51,52]. Three symplastic barriers are therefore defined: (1) inner/outer integuments, (2) inner integuments/endosperm, (3) endosperm/embryo. Transport across those symplastic barriers necessitates a carrier-mediated transport, such as the sucrose transporters (SUC or SUT) or the ABCB/PGP and PIN transporters. Moreover, the presence of these symplastic barriers between the sporophytic seed coat, the triploid endosperm and the diploid embryo prevents symplastic transport to be considered as communication means for coordinated development between those three structures.

The symplastic barrier between the embryo and the endosperm is formed after deposition of a cuticle outside the proembryo around globular stage [53]. Because of the absence of a cuticle between the endosperm and the suspensor, the suspensor cells may be involved in a nutrient uptake via transmembrane transport [50]. Later during the development, suspensor cells undergo programmed cell death [54]. Therefore, after the torpedo stage, during seed maturation, uptake of nutrients may occur directly through the embryo epidermis from the endosperm, despite the presence of the cuticle membrane [54]. Outside the cuticle, at the embryo surface, is deposited an embryo sheath [55]. Involved in the organization and the integrity of the cuticle (but not its biogenesis) are peptides secreted by the endosperm: ESR-specific subtilisin protease ALE1 (ABNORMAL LEAF-SHAPE1) and embryonic epidermis-expressed receptor kinases GASSHO1 (GSO1) and GSO2 [53,55]. The genes involved in cuticle biogenesis are expressed as early as the mid-globular stage, after the formation of the protoderm [53]. The expression of these genes is GSO1- and GSO2-independent. The formation of the extra-cuticular sheath requires a signaling pathway involving the endosperm-produced peptide KERBEROS (KRS) in parallel of the ALE1/GSO1/GSO2 pathway [55]. Moreover, the endosperm-specific transcription factor ZHOUPI (ZOU) regulates the expression in the endosperm of both *ALE1* and *KRS* [55]. The cuticle is initially deposited as patches to form a uniform layer later, resulting in the isolation of the embryo from its surrounding endosperm from early torpedo stage. As deduced from the *krs* mutant phenotypes the sheath appears to work as a lubricant for the growing movement of the embryo into the cellularized endosperm [55]. In an attempt to uncover downstream components of the ALE1-GSO1-GSO2 pathway, a transcriptomic approach identified a link of this pathway to genes involved in plant defence responses. By analogy, the authors studied the MAP-kinase MPK6 as a possible downstream embryonic component of the signaling pathway [53].

Symplastic transport and symplastic barriers create domains within the seed structures where molecular transport is facilitated. Furthermore, the cuticle is a clear example of endosperm-embryo cooperation for the formation of a symplastic barrier between the two structures (Figure 2). Further studies would determine the role of this communication means for the growth coordination of the embryo, endosperm and seed coat.

## 4. Movement of Sucrose within the Seed, Another Example of Molecular Communication

Sucrose is a necessary nutrient for embryo and seed development. Inflorescences, and seeds, in particular, are major carbon sinks from sucrose produced in mature leaves [59,60]. Sucrose is a sugar transported on long distance from maternal tissues, via the funicular phloem, through the seed coat and the endosperm to the embryo (Figure 2). Feeding experiment in oilseed rape identified the flow of sucrose in seeds from the integuments to the embryo via the ESR, and to the central endosperm via the chalazal endosperm [56]. It requires the help of plasma membrane sucrose transporters (SUT or SUC) and sucrose exporter (SWEET) [61,62]. In the endosperm, the sucrose is converted into hexoses (glucose and fructose) [56]. The hexose/sucrose ratio is high in the endosperm and will influence the sugar movement from maternal tissues to the seed coat and ultimately to endosperm and embryo. This sucrose flow to the embryo is crucial for seed size and viability [54,59]. A reduction in the maternal sucrose flow to the embryo by mutations in the *SWEET*s and (endosperm expressed) *AtSUC5* induces a delay in the embryo development and a reduction in embryo and seed size, and seed weight [57,58]. The hexose concentration gradient between endosperm and embryo is in favor of endosperm nuclear division over zygote division in the initial developmental phase following fertilization [63,64]. Evidence of crosstalk between auxin and sucrose in the balance between cell elongation and cell division for seed development have been discussed by Wang and Ruan (2013) [64].

Endosperm cellularization starts from micropyle progressing towards the chalazal region after the 8th mitotic division (heart stage). This cellularization reduces the size of the endosperm vacuole, main storage pool for hexoses in the seed [56]. Therefore, a decrease of the vacuole size by cellularization decreases hexose levels in the endosperm, and thus the sink strength between the embryo and the endosperm. Indeed, endosperm cellularization correlates with a decrease in the hexose/sucrose ratio [65]. Several mutants affected in endosperm cellularization also display an arrest of embryonic development [66]. This is the case of *fis2*. This repressive FIS-PRC2 complex is essential for suppressing seed development in the absence of fertilization, as discussed above. Loss of *FIS2* causes endosperm proliferation and seed abortion. In *fis2*, the absence of endosperm cellularization results in high hexose levels and an absence of sucrose transfer to the embryo, leading to its developmental arrest. The absence of *AGL62*, downstream signaling component of the FIS2 pathway, leads to early endosperm cellularization. Indeed, reducing the expression levels of *AGL62* in the endosperm rescues *fis2* endosperm phenotypes. In the *fis2 agl62* double mutant, embryo development is delayed compared to wild-type embryos, correlating with a delayed endosperm cellularization and a reduced hexose content compared to *fis2* seeds [66]. The *fis2* endosperm phenotypes show the importance of endosperm cellularization for its role as a nutrient (hexose) provider for embryo growth. This is supported by the retarded embryonic growth observed in endosperm-expressed *sweet* mutants, impaired in sucrose transport from the endosperm to the embryo [57,66]. No direct links were so far identified between SWEET and AGL62. However, the function of FIS-PRC2 and AGL62 signaling pathway during endosperm development associates post-fertilization endosperm initiation and auxin production (for seed coat differentiation) with endosperm cellularization and endosperm-to-embryo flow of sucrose (for embryo growth).

## 5. Seed to Fruit Communication: Hormonal Crosstalk for Fruit Initiation

Coordinated seed growth alone is not enough for successful reproduction. It requires a synchronized growth with the fruit (or silique). For example, ectopic auxin production in the central cell induces fruit parthenocarpy and links ovule fertilization to fruit development [15]. Several lines of evidence suggest that fertilization-dependent auxin production and signaling in the seeds activates GA production and signaling for fruit initiation [1,67,68,69,70]. This fertilization-dependent crosstalk is mimicked by auxin application, whereas GA application does not have any effects on auxin signaling [15]. Additionally, auxin effects on fruit initiation require unimpaired GA production and signaling pathways in *Arabidopsis* [70,71]. Expression analysis by RT-qPCR performed on dissected carpel valves and in ovules/seeds at anthesis and post-anthesis in unpollinated and pollinated pistils demonstrated that genes involved in GA biosynthesis are upregulated exclusively in the fertilized ovules, but not in the carpels. Validating this observation is the upregulation of GA biosynthetic genes in auxin-treated unpollinated pistils [1]. The binding of GA to its receptor, GID1 (GIBBERELLIN INSENSITIVE DWARF1) activates a GA signaling pathway involving the degradation of the transcriptional repressor DELLA proteins [72,73]. Mutant combinations in multiple DELLA proteins mimic an excess of GA, auto-activate GA responses and produce parthenocarpic fruits [70]. The DELLA protein RGA (REPRESSOR OF GA1–3) is mainly present in ovules, and weakly in carpel valves. The degradation of the RGA signals in both tissues by pollination or auxin treatments implies the activation of the GA signaling pathway in both ovules and carpel valves [1]. Alterations in GA signaling or production in tomato and *Arabidopsis* trigger similar results [1,70,74]. In tomato, direct crosstalk between auxin and GA has been uncovered with the interaction of SlARF7 with both SlDELLA and SlIAA9 [74]. This dual interaction antagonistically regulates the expression of genes involved in GA and auxin metabolism and synergistically regulates the expression of genes involved in growth, such as *EXPANSIN5* and the ethylene production gene *ACC OXIDASE4* [74]. In *Arabidopsis*, tomato and eggplant, downregulation of ARF7 and ARF8 proteins, by RNA interference or loss-of-function mutations, results in parthenocarpic fruits [74,75,76,77], suggesting that the IAA9/ARF7 auxin signaling pathway represses fruit growth in the absence of ovules fertilization by repressing *EXPANSIN5* expression. GA (or GA-derived signal) is, therefore, a good candidate for a molecular communication component between fertilized ovules and carpel valves to promote and coordinate fruit growth and seed development [1].

## 6. Conclusions

In this review, I discussed recent advancements in our knowledge on inter-tissular molecular communication between the three seed structures (seed coat, endosperm, embryo) and the fruit. The seed is compartmentalized in symplastic domains, that isolate the three seed structures from each other’s and impose a controlled transmembrane transport for communication between two symplastic domains. I emphasize the relevant role of auxin as a signaling molecule and its regulated transport between these three structures. Also, sucrose, as nutrients, is another example of small molecules involved in the communication between the seed structures. Altogether, hormones and nutrients are critical molecular components for the coordination of growth and development of the seeds and the fruit.

## Figures and Tables

**Figure 1 ijms-20-00936-f001:**
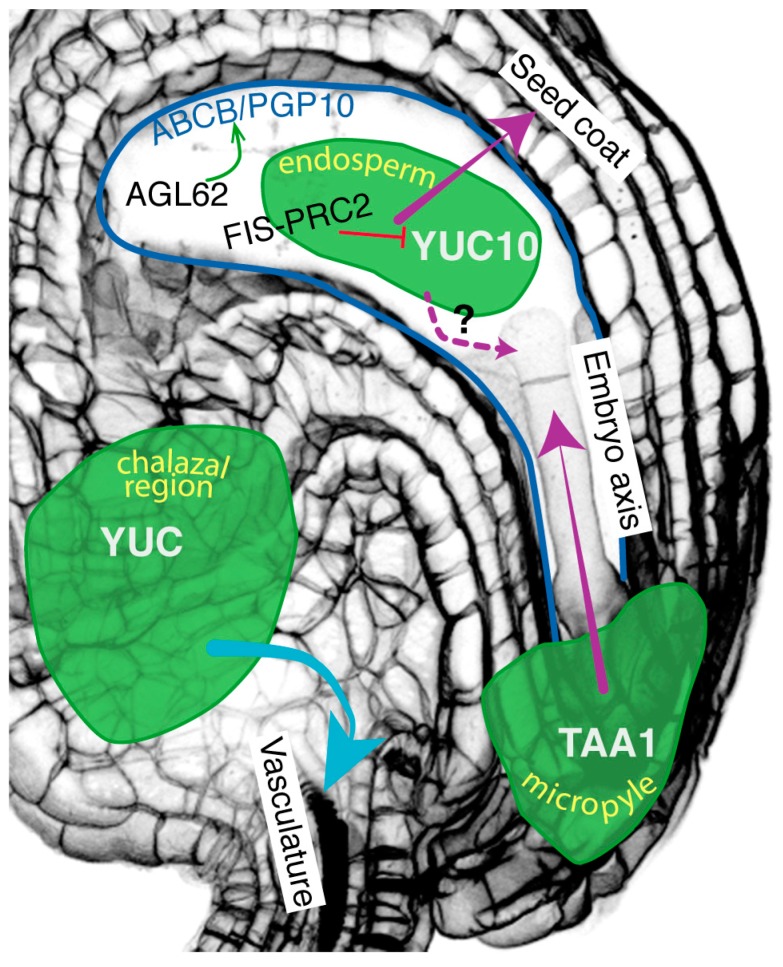
Auxin movement between the three seed structures after fertilization. Auxin production sites are symbolized in green. Auxin in the chalazal region promotes the specification of the vasculature in the funiculus (auxin sporophytic movement symbolized by the blue arrow). Auxin at the micropyle moves to the embryo to contribute to the formation of the embryo body axis. Auxin production in the endosperm is epigenetically repressed before fertilization by FIS-PRC2 (FERTILIZATION INDEPENDENT SEED-Polycomb Repressive Complex 2) complex (red line). This auxin moves to the integuments to contribute for the differentiation of the seed coat, through ABCB/PGP10 auxin efflux proteins. Auxin movement between different seed structures are symbolized in purple (endosperm-integuments, integuments-embryo). The expression of *ABCB/PGP10* is dependent on AGL62 (green arrow). No regulated auxin movement between the endosperm and the embryo has been yet demonstrated (dotted purple arrow).

**Figure 2 ijms-20-00936-f002:**
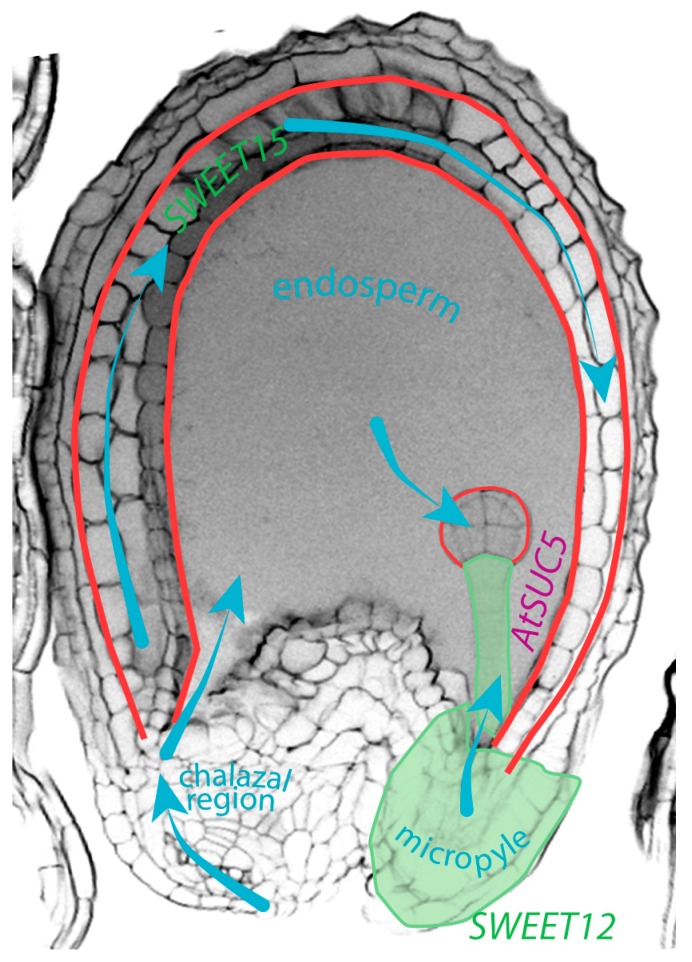
Symplastic barriers in the seed are illustrated (red): inner/outer integuments, inner integuments/endosperm, endosperm/embryo (cuticle). Blue arrows indicated the sucrose flow in the seed, deduced from feeding experiments in oilseed rape [56]. The expression pattern of *SWEET12* (micropyle and suspensor) and *SWEET15* (seed coat) is indicated in green [57]. The expression pattern of *AtSUC5* in ESR is shown in purple [58].

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
