# Peer review of "Molecular Communication for Coordinated Seed and Fruit Development: What Can We Learn from Auxin and Sugars?"

_ijms, 2019, doi:10.3390/ijms20040936_

Reviewer 1 Report

In this manuscript, Hélène Robert reviews the current understanding of how the different compartments of the seed communicate with each other (embryo, endosperm and seed coat) and with the fruit. For this, the author explores the dynamics of the hormone auxin and sugar transport during seed development, both of which have been extensively characterized. The author then uses this information to dissect the possible intercellular communication routes that are available to the seed structures, with a focus on the model plant Arabidopsis thaliana.

This is an interesting review manuscript because it offers an unexplored and updated perspective on which are the possible communications pathways between embryo, endosperm and seed coat. I only have minor details that should be addressed prior to publication:

- Throughout the manuscript gene names and mutant names are not italicized. This may have been a formatting issue, but should be corrected.

- Line 67 - should read "which elongates and divides asymmetrically" ("divides" is missing).

- Line 142 - should be "prior to fertilizaton" ("to" is missing).

- Line 165 - the gene should be YUC6 and not YUC10.

- Line 258 (legend of Fig. 1) - should be "contribute" instead of "contribution".

- Figure 2 - The quality of this figure is low compared to figure 1, which makes it hard to see the arrows (could have been just an issue with importing the figure).

Author Response

I would like to thank the reviewer for her/his positive words on the manuscript. I have corrected the revised text accordingly to the comments.

- Throughout the manuscript gene names and mutant names are not italicized. This may have been a formatting issue but should be corrected.

It might have been indeed a formatting issue as in the original text, the names were italicized. I have corrected it in the revised text and I will double-check the manuscript after upload.

- Line 67 - should read "which elongates and divides asymmetrically" ("divides" is missing).

- Line 142 - should be "prior to fertilization" ("to" is missing).

- Line 165 - the gene should be YUC6 and not YUC10.

- Line 258 (legend of Fig. 1) - should be "contribute" instead of "contribution".

These mistakes have been corrected. 

- Figure 2 - The quality of this figure is low compared to figure 1, which makes it hard to see the arrows (could have been just an issue with importing the figure).

Figure 2 quality was improved. 

Reviewer 2 Report

The review written by Helene Roberts present an up-to-date knowledge about the coordination mechanisms between the embryo and endosperm, within a developing seed and in context of flower development.

 The review is very well structured and provide all the known facts (as far as I know) in a logical scheme. I was glad to read this.

 The only disadvantage I see, is that the sugars topic is not described similarly well as auxin topic. Only one chapter 4 is fully related to sugars the info about sugars in other chapters is very scarce. Maybe the author will succeed in adding more sucrose-related information into the chapters 2,3 and 5. E. g. the final sentence of the chapter 1 (line 90-91) says that the whole story will be about auxin, no any word about sugars.

Maybe the figure 2 might be updated with the information about the sucrose transporters.

 I also suggest to improve the figure 1 and add subscriptions for micropyle, embryo, chalazal region.

 The names of all the mutants and all the genes must be done in italics

 Lines 242 – 253 Should be said that it is a summary of written above.

Reference to the figure 2 should be given earlier (e.g. line 279).

Line 287 uptake instead of update.

Author Response

I thank the reviewer for her/his positive comments on the manuscript. Please find below point-to-point reply to the comments.

- The only disadvantage I see is that the sugars topic is not described similarly well as auxin topic. Only one chapter 4 is fully related to sugars the info about sugars in other chapters is very scarce. Maybe the author will succeed in adding more sucrose-related information into the chapters 2,3 and 5. E. g. the final sentence of chapter 1 (line 90-91) says that the whole story will be about auxin, no any word about sugars.

To my knowledge, detailed molecular information about sugar transport within the seed structures is rather scarce compared to what is known on auxin. However, I added Chapter 4 in the review to illustrate that we should not be blind on the only role of auxin in this topic and other molecules such as nutrients may be involved in inter-tissular communication to growth coordination. In agreement with the editor, I added a sentence at the end of Chapter 1 to introduce the chapter on sucrose:

"In chapter 4, I will additionally shortly discuss the role of a second molecule, sucrose, as a mean of coordinating the development of seed structures." lines 91-92.

- Maybe the figure 2 might be updated with the information about the sucrose transporters.

I also suggest to improve the figure 1 and add subscriptions for micropyle, embryo, chalazal region.

 Both figures have been modified accordingly. 

- The names of all the mutants and all the genes must be done in italics

 It appears that during the formatting and the upload to the ijms system of the original manuscript, the italics disappeared. I have corrected it in the revised text and will double-check the manuscript after upload. 

- Lines 242 – 253 Should be said that it is a summary of written above.

"In summary" has been added to start the paragraph.

- Reference to figure 2 should be given earlier (e.g. line 279).

A reference to Figure 2 has been included in line 275, after "There are several symplastic domains identified in seeds".

- Line 287 uptake instead of update.

This mistake has been corrected.